# Changing Epidemiological Trends of Hepatobiliary Carcinomas in Austria 2010–2018

**DOI:** 10.3390/cancers14133093

**Published:** 2022-06-23

**Authors:** Florian Hucke, Matthias Pinter, Miriam Hucke, Simona Bota, Dajana Bolf, Monika Hackl, Markus Peck-Radosavljevic

**Affiliations:** 1Internal Medicine and Gastroenterology (IMuG), Hepatology, Endocrinology, Rheumatology and Nephrology Including Centralized Emergency Department (ZAE), Klinikum Klagenfurt am Wörthersee, 9020 Klagenfurt, Austria; miriam.hucke@kabeg.at (M.H.); simona.bota@kabeg.at (S.B.); dajana.bolf@kabeg.at (D.B.); markus.peck-radosavljevic@kabeg.at (M.P.-R.); 2Division of Gastroenterology and Hepatology, Department of Internal Medicine III, Liver Cancer (HCC) Study Group Vienna, Medical University of Vienna, 1090 Vienna, Austria; matthias.pinter@meduniwien.ac.at; 3Austrian National Cancer Registry, Statistics Austria, 1110 Vienna, Austria; monika.hackl@statistik.gv.at

**Keywords:** hepatocellular carcinoma, biliary tract cancers, cholangiocarcinoma, gallbladder carcinoma, ampullary carcinoma, incidence, mortality, survival

## Abstract

**Simple Summary:**

Primary liver cancer is currently the sixth most common cancer and the third common cause of cancer-related mortality worldwide. Incidences have increased in recent year, especially in high-income countries. The epidemiology of predisposing risk factors for hepatobiliary carcinomas have changed significantly, while treatment and therapeutic options have markedly improved. Here, we provide an update of incidence, mortality, and survival trends in recent years, in Austria. While age-adjusted incidence rates remained stable in almost all hepatobiliary carcinoma subtypes—except for gall-bladder cancer—the overall survival improved significantly.

**Abstract:**

Using national registries, we investigated the epidemiological trends of hepatobiliary carcinomas in Austria between 2010 and 2018 and compared them to those reported for the periods of 1990–1999 and 2000–2009. In total, 12,577 patients diagnosed with hepatocellular carcinoma (*n* = 7146), intrahepatic cholangiocarcinoma (*n* = 1858), extrahepatic cholangiocarcinoma (*n* = 1649), gallbladder carcinoma (*n* = 1365), and ampullary carcinoma (*n* = 559), between 2010 and 2018, were included. The median overall survival of all patients was 9.0 months. The best median overall survival was observed in patients with ampullary carcinoma (28.5 months) and the worst median overall survival was observed in patients with intrahepatic carcinoma (5.6 months). The overall survival significantly improved in all entities over the period 2010–2018 as compared with over the periods of 2000–2009 and 1990–1999. Age-adjusted incidence and mortality rates remained stable for most entities in both, men and women; only in gallbladder carcinoma, the incidence and mortality rates significantly decreased in women, whereas, in men, the incidence rates remained stable and mortality rates showed a decreasing trend. We showed that age-adjusted incidence and mortality rates were stable in most entities, except in gallbladder carcinoma. The overall survival improved in almost all entities as compared with those during 1990–2009.

## 1. Introduction

Hepatobiliary carcinomas (HBC) include hepatocellular carcinoma (HCC) as well as biliary tract cancers (BTCs). BTCs are invasive adenocarcinomas, which comprise, corresponding to their site of origin, intrahepatic cholangiocarcinoma (iCC), extrahepatic cholangiocarcinoma (eCC), as well as gallbladder carcinoma (GBC), and cancer of the ampulla of Vater (ampullary carcinoma, AC). To date, primary liver cancer, with HCC (75%) and iCC (15%) as the main subtypes, represents the sixth most frequent type of cancer worldwide and the third most common cause of cancer-related mortality [1,2].

Interestingly, incidences of the HBC subtypes vary significantly across geographical regions and show a strong relation with the sociodemiographic index. While incidences of HCC and iCC are rising mostly in high-income countries, those of eCC, AC, and GBC have been reported to remain stable or even decrease [3].

Changes in the incidence of the various HBC subtypes may reflect trends in socioeconomic burden and life-style behavior. In addition to classical risk factors such as primary sclerosing cholangitis and Caroli’s disease for cholangiocarcinomas or cholelithiasis (gallstones) for GBC, lifestyle factors, including excessive alcohol consumption or smoking and obesity- and diabetes-related diseases such as non-alcoholic fatty liver disease (NAFLD) or non-alcoholic steatohepatis (NASH), are now considered to be major risk factors for the development of HBC. The increasing emergence of these lifestyle associated factors, the augmented availability of diagnostic means, and the enhanced access to treatment options for liver inflammation, cirrhosis, and viral hepatitis have modified the relative distribution of the risk factors and have contributed to the increasing incidence rates of HCC and cholangiocarcinoma in high-income regions [1,4,5,6]. Despite considerable improvements of HBC treatment in recent years, especially in terms of systemic therapies (e.g., targeted therapies and immunotherapy) [1,6], most patients (>60%) with HCC [7] or BTC [1,6,8] are still diagnosed at intermediate or even advanced tumor stages, when curative treatment is no longer feasible.

Here, we aimed to investigate the impact of the changing landscape of risk factors and of earlier diagnostic approaches (e.g., surveillance recommendations [9] and routine cholecystectomy) together with improvements in therapeutic options in high-income countries on the epidemiology of HBC for the period 2010–2018. This is a follow-up to our previous reports on incidence and mortality trends of HCC [10] and BTC [11] in Austria between 1990 and 2009.

## 2. Material and Methods

### 2.1. Data Retrieval

National electronic databases were used to analyze epidemiological trends of Austrian patients diagnosed with hepatobiliary tumors between January 2009 and December 2018.

Data on incidence (last updated in December 2021) were obtained from the Austrian National Cancer Registry (ANCR), which compiles data on all newly diagnosed cases of cancer in Austria, a country of 8.8 million (2018) inhabitants. All patients with hepatobiliary tumors in Austria are registered at the ANCR as notification is obliged by law (Cancer Statistics Act 1969 and Cancer Statistics Ordinance 2019). This is a population-based registry and has been operated by the National Statistical Institution (Statistics Austria) since 1969. Regional cancer registries exist in four out of nine federal provinces (Vorarlberg, Tyrol, Salzburg, and Carinthia), which report their cancer cases to Statistics Austria. In the other federal provinces, data on cancer cases are reported directly to Statistics Austria by local hospitals. For follow-up and to ascertain death certificate only (DCO) cases, ANCR data are linked with the official causes of death (CoD) statistics derived from Statistics Austria since 1983.

The collected information in this study included general demographic characteristics of the patients (age and sex); tumor site; histological type according to the International Classification of Diseases for Oncology (ICD-O-3), which was transformed to the International Statistical Classification of Diseases and Related Health Problems (ICD-10); and the stage of tumor. Furthermore, the year of diagnosis, the year of death and the survival time in days was obtained from Statistics Austria. Follow-up until 31 December 2019 was based on cancer notification forms and data linkage with the CoD Statistics. The data were subdivided into the following six groups according to the four-digit code of the International Classification of Diseases, 10th revision (ICD10):

(1) hepatocellular carcinoma (C22.0); (2) intrahepatic cholangiocarcinoma (C22.1); (3) extrahepatic cholangiocarcinoma (C24.0); (4) cancer of the ampulla of Vater (C24.1); (5) gallbladder carcinoma (C23); (6) other tumors including overlapping lesion of the biliary tract (C24.8) and biliary tract, unspecified (C24.9). Group 6 was a priori excluded and not further analyzed.

Furthermore, data from our previous publications (time period 1999–2009) [10,11] were included in this study, which were obtained from the same sources as described above in 2013. To analyze trends in overall survival (OS) over time, three time periods were defined: 1990–1999, 2000–2009, and 2010–2018.

### 2.2. Statistics

Baseline characteristics were summarized using descriptive statistics. An χ^2^-test was used to compare categorical data. Parametric data were compared using Student’s *t*-test. Age-standardized incidence and mortality rates (per 100,000—European Standard Population, 2013) were calculated by year of diagnosis and sex. Time trends in incidence and mortality were assessed by bivariate Spearman correlation; sex differences were calculated by the unpaired Mann–Whitney U test.

Survival was defined as the time from date of diagnosis to date of death or last follow-up (31 December 2019). DCO cases were excluded from the survival analyses. OS was calculated using the Kaplan–Meier method and compared by means of the log rank test. A *p*-value <0.05 was considered to be significant. Statistical analyses were performed using the statistical software package IBM SPSS version 25.0 (IBM Corp., Armonk, NY, USA).

## 3. Results

### 3.1. Patients

In total, 12,577 patients diagnosed with HCC (*n* = 7146), iCC (*n* = 1858), eCC (*n* = 1649), GBC (*n* = 1365), and AC (*n* = 559) between 2010 and 2018 were included in this analysis; 311 (2%) patients had unspecified (ICD-10 code C24.9) or overlapping tumors (ICD-10 code C24.8) and were not further analyzed as mentioned above (Appendix A).

Detailed patient characteristics are shown in Table 1. The mean age at diagnosis was 71 years and 63% of patients were male. Distant or lymph node metastases were present in 12% and 10% of cases, respectively.

There were 9959 subjects (79%) who were available for survival analyses, 23% of the subjects were DCO cases and excluded. The median OS was 9.0 months (95% confidence interval (CI) 8.6–9.4) with 1- and 5-year survival rates of 44% and 8%, respectively.

### 3.2. Hepatocellular Carcinoma (HCC, ICD-10 code C22.0)

Among 7146 patients diagnosed with HCC, 5342 patients (75%) were male and 1804 patients (25%) were female. The mean age at diagnosis was 70 years. Distant or lymph node metastases were present in 8% and 4%, respectively (Table 1).

Between 2010 and 2018, the age-adjusted incidence rate was five times higher in men than in women (*p* < 0.001) (Figure 1a). The incidence rate remained stable in men (range 11.9–13.4, *p* = 0.898), whereas in women a trend towards a decreasing incidence was observed (from 2.7 in 2010 to 2.0 in 2018, *p* = 0.067).

The age-adjusted mortality was 4.9 times higher in men than in women (*p* < 0.001, Figure 1b). Mortality rates remained unchanged in both, men (range 8.6–11.1, *p* = 0.668) and women (range 1.6–2.5, *p* = 0.898), between 2010 and 2018.

A total of 5574 (78%) patients were available for survival analysis, 1572 (22%) were DCO cases, and therefore, had to be excluded from the analysis. The median OS of patients diagnosed with HCC between 2010 and 2018 was 9.3 months (95% CI 8.6–10.0, Figure 6). Men had a slightly better OS (9.5 months, 95% CI 8.7–10.4) than women (6.8 months, 95% CI 7.6–9.8, *p* = 0.050, Figure 1c). The 1- and 5-year survival rates in men and women were 46/43% and 8/8%, respectively.

As compared with the historical cohorts of patients diagnosed with HCC in 1990–1999 (2.6 months, 95% CI 2.3–2.9) and in 2000–2009 (5.6 months, 95% CI 5.1–6.1), the OS improved significantly in the time period 2010–2018 (*p* < 0.001, Figure 1d), while the 1-year survival rate (25 vs. 37 vs. 46%) improved and the 5-year survival rate remained stable (8 vs. 7 vs. 8%).

### 3.3. Intrahepatic Cholangiocarcinoma (iCC, ICD-10 code C22.1)

With 15% of all HBC, iCC was the second most common tumor. Among 1858 patients, 997 patients (54%) were male and 861 patients (46%) were female. The mean age at diagnosis was 71 years. Distant or lymph node metastases were present in 22% and 9%, respectively (Table 1).

The age-adjusted incidence rate was significantly higher in men than in women (m/f ratio 1.6:1, *p* < 0.001). The incidence rates remained stable in men (range of 3.4–5.8, *p* = 0.460) and women (range of 2.0–3.5, *p* = 0.154) (Figure 2a).

The age-adjusted mortality was 1.6 times higher in men as compared with women (*p* < 0.001). The mortality rates remained stable in both, men (range of 3.8–4.9, *p* = 0.286) and women (range of 1.9–3.2, *p* = 0.139) (Figure 2b).

Seventy-six percent of patients were available for survival analyses; 24% of the patients were DCO cases, and therefore excluded. The median OS of patients diagnosed with iCC between 2010 and 2018 was 5.6 months (95% CI 5.0–6.3, Figure 6). The OS was almost equal in men (5.7 months, 95% CI 4.7–6.7) and women (5.5 months, 95% CI 4.7–6.3, *p* = 0.486) (Figure 2c). The 1- and 5-year survival rates in men and women were 32/32% and 4/5%, respectively.

The median OS was significantly better (*p* < 0.001, Figure 2d) in 2010–2018 as compared with in 1990–1999 (3.5 months, 95% CI 3.0–3.9) or in 2000–2009 (5.3 months, 95% CI 4.8–5.9). There was an improvement in the 1-year survival rate (22 vs. 30 vs. 32%), but not in 5-year survival rate (5 vs. 3 vs. 4%).

### 3.4. Extrahepatic Cholangiocarcinoma (eCC, ICD-10 code C24.0)

Among 1649 patients diagnosed with eCC, 884 patients (54%) were male and 765 patients (46%) were female. The mean age at diagnosis was 73 years. Distant or lymph node metastases were observed in 13% and 18%, respectively (Table 1).

The age-adjusted incidence rate was significantly higher in males than in females (m/f ratio 1.6:1, *p* < 0.001). The age-adjusted incidence rate remained stable over time in men (range 2.2–3.3, *p* = 0.637) and women (range 1.3–2.4, *p* = 0.308) (Figure 3a). The age-adjusted mortality rate was 1.4 times higher in men than in women (*p* < 0.001). There was no significant trend in the age-adjusted mortality rates in men (range 1.3–2.9, *p* = 0.224) or in women over time (range 0.9–1.8, *p* = 0.099) (Figure 3b).

Eighty-two percent of patients were available for survival analyses, while 18% of patients were DCO cases and as such excluded from survival analyses. The median OS of patients diagnosed with eCC was 9.1 months (95% CI 8.2–10.1, Figure 6). The OS was significantly longer in men (10.1 months, 95% CI 9.5–12.5) than in women (7.1 months, 95% CI 6.0–8.3, *p* = 0.001) (Figure 3c). The 1- and 5-year survival rates in men and women were 48/37% and 6/5%, respectively.

As compared with the time periods 1990–1999 (4.4 months, 95% CI 3.8–5.1) and 2000–2009 (6.3 months, 95% CI 5.3–7.2), the OS was significantly better in the current time period between 2010 and 2018 (*p* < 0.001, Figure 3d). The corresponding 1-year survival rate (29 vs. 35 vs. 43%) improved significantly, however, without any changes in the 5-year survival rate (7 vs. 4 vs. 6%).

### 3.5. Gallbladder Carcinoma (GBC, ICD-10 code C23.0)

Among 1365 patients diagnosed with GBC, 437 patients (32%) were male and 928 patients (68%) were female. The mean age at diagnosis was 71 years. Distant or lymph node metastases were observed in 25% and 19%, respectively (Table 1).

The age-adjusted incidence rate was significantly higher in women than in men (m/f ratio 1:1.6, *p* < 0.001). While the age-adjusted incidence rates remained stable in men (range 1.1–1.6, *p* = 0.356), there was a significant decrease over time in women from 3.0 in 2010 to 1.9 in 2018 (*p* = 0.002) (Figure 4a).

The age-adjusted mortality rates were 1.8 times higher in women than in men (*p* < 0.001) and decreased significantly over time in women (range 1.2–2.2, *p* = 0.010). Age-adjusted mortality rates showed a decreasing trend in men (from 1.1 to 0.6) over time, however, this was not statistically significant (*p* = 0.067) (Figure 4b).

Eighty-two percent of patients were available for survival analyses, while 18% of patients were DCO cases, and therefore excluded from the analysis. The median OS was 8.5 months (95% CI 7.5–9.4, Figure 6), with no significant difference between men (8.8 months, 95% CI 7.3–10.4) and women (8.2 months, 95% CI 7.0–9.3, *p* = 0.854, Figure 4c). The 1- and 5-year survival rates for men and women were 42/40% and 7/8%, respectively.

As compared with the historical time periods 1990–1999 (3.4 months, 95% CI 3.1–3.7) and 2000–2009 (4.9 months, 95% CI 4.3–5.4), there was a significant improvement of survival over time between 2010 and 2018 (*p* < 0.001, Figure 4d), while the 1-year survival rate (25 vs. 32 vs. 41%) improved, and the 5-year survival rate remained stable (10 vs. 6 vs. 8%).

### 3.6. Carcinoma of the Ampulla of Vater (AC, ICD-10 code C24.1)

Among 559 patients diagnosed with AC, 299 patients (54%) were male and 260 patients (46%) were female. The mean age at diagnosis was 71 years. Distant or lymph node metastases were observed in 8% and 30%, respectively (Table 1).

The age-adjusted incidence rate was 1.5 times higher in men than in women (*p* = 0.001) and fluctuated between 0.7 and 1.4 in men (*p* = 0.332) and between 0.4 and 0.9 in women (*p* = 0.433) (Figure 5a). The age-adjusted mortality rate was 1.3 times higher in men than in women (*p* = 0.019) and fluctuated between 0.2 and 0.6 in men (*p* = 0.637) and between 0.2 and 0.5 in women (*p* = 0.831) (Figure 5b) over time.

Ninety-one percent of cases were available for survival analysis, while 9% of cases were DCO cases, and therefore excluded from the analysis. The median OS was 28.5 months (95% CI 22.6–34.4, Figure 6) with no statistical difference between men (28.5 months, 95% CI 22.5–34.5) and women (27.5 months, 95% CI 16.1–38.8, *p* = 0.720) (Figure 5c). The 1- and 5-year survival rates in men and women were 70/66% and 16/19%, respectively.

The OS in 2010–2018 was better than in 1990–1999 (18.3 months; 95% CI: 14.6–21.8) and in 2000–2009 (18.2 months; 95% CI: 14.5–21.8, *p* = 0.003; Figure 5d). The corresponding 1-year survival rate (60 vs. 60. vs. 68%) improved, while the 5-year survival rate (25 vs. 14 vs. 18%) remained unchanged.

**Figure 6 cancers-14-03093-f006:**
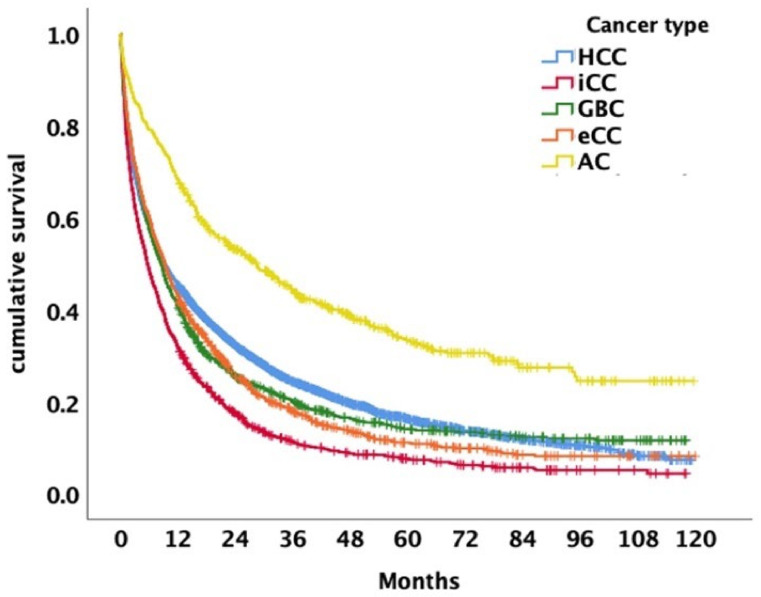
Kaplan–Meier curves of patients diagnosed with biliary tract cancer: hepatocellular carcinoma (HCC) vs. intrahepatic cholangiocarcinoma (iCC) vs. gallbladder carcinoma (GBC) vs. extrahepatic cholangiocarcinoma (eCC) vs. ampullary carcinoma (AC) (median OS 9.3 vs. 5.6 vs. 8.5 vs. 9.1 vs. 28.5 months, *p* < 0.001). Pooled median OS (*n* = 9959) 9.0 months (95% CI 8.6–9.4) and 1-/5-year OS, 44%/8%.

## 4. Discussion

Here, we show that, despite unchanged age-adjusted incidence and mortality rates of HCC, iCC, eCC, and AC over time, the median OS of all HBC subtypes was significantly higher in 2010–2018 as compared with 1990–1999 and 2000–2009. Only GBC showed decreasing incidence and mortality rates, which were statistically significant in women but showed only a trend towards improvement in men.

HCC is the most common primary liver cancer (75%), which mainly occurs in patients with underlying chronic liver disease. In Europe [12], the incidence of HCC is rising, with the exception of Italy where a continuously decreasing trend has been observed. Incidence [13] and mortality [14] rates have both been predicted to steadily increase in Austria. We could not confirm those predicted trends in our present work, since incidence and mortality rates remained stable between 2010 and 2018. This could probably be explained by the changing epidemiology of underlying chronic liver diseases. Excessive alcohol consumption is associated with up to 16% higher risk of developing HCC and is the most common etiological factor in HCC patients in Austria. Recently, a continuous reduction in alcohol consumption has been observed in Austria [15].

The second most common risk factor for HCC in Austria and the most common etiology in Western countries is chronic hepatitis C virus (HCV) infection. Recently, a major breakthrough in HCV therapy was achieved by the development of direct-acting antivirals (DAAs) [16]. Achieving a sustained virological response by DAA therapy reduces the risk of developing HCC by 50–80% [17,18]. In contrast to other countries, treatment with DAAs has been quickly and easily accessible to all infected patients in Austria, leading to higher treatment rates as compared with the interferon era. This might have prevented a larger number of subsequent HCV-associated HCC cases. In 2016, the first global health sector strategy (GHSS) on chronic viral hepatitis was published by the World Health Organization (WHO). Two targets were to reduce new hepatitis infections by 90% and related deaths by 65% until 2030 [19]. Despite high treatment rates, achieving those goals will still need more effort, especially in identifying HCV-infected patients outside of the classical risk groups [20]. The COVID-19 pandemic may have had a significant deteriorating effect on the achievement of these goals, since the initiation of HCV treatment was considerably hampered or even stopped in the year 2020. According to calculation models, a one-year delay in HCV programs will lead to an additional 44,800 HCC cases worldwide [21]. In contrast to HCV, hepatitis B virus infections play only a negligible role in the development of HCC in Western Europe [5] and in Austria due its rather low prevalence.

While viral hepatitis and alcohol-associated HCC incidences have declined, the overall HCC incidence continues to increase in high-income countries. This effect is mainly caused by a changing trend in risk factors. The prevalences of diabetes, obesity, and hypertension are rising continuously, leading to higher rates of NAFLD and NASH. Worldwide, more than 200 million people are currently affected by NAFLD [22]. In the USA, the prevalence of NAFLD amounts to approximately 32.8% [23]. The risk of progression from NAFLD to NASH is estimated to be 20%, of which 10–20% continue to progress to cirrhosis. About 20–30% of NASH-induced HCCs develop even in the absence of cirrhosis [24]. Currently, NAFLD is already one of the leading risk factors for cirrhosis and HCC in the USA. Although our observations suggested that NASH was an upcoming risk factor for HCC in Austria even 10 years ago [25], we could not confirm a definite increase in NAFLD-associated HCC rates in our daily clinical practice to date.

Despite comparable incidence and mortality rates, we observed a significant amelioration of the OS over the last three decades. A major reason might be the tighter adherence to international treatment guidelines. Endorsed by the European Association for the Study of Liver Disease (EASL) and American Association for the Study of Liver Disease (AASLD) the implementation of the Barcelona Clinic Liver Cancer (BCLC)-treatment algorithm [9,26,27] into clinical practice in most countries standardized and markedly improved the management of patients with HCC. This staging system and treatment algorithm links tumor stages to recommended first-line therapies and has been updated several times; the last update was in 2022 [28]. Furthermore, recommendations for HCC surveillance in patients with liver cirrhosis has led to an earlier diagnosis of HCC [29] and might explain the attenuated rates of distant or lymph node metastases we observed in our cohort over the last three decades. However, the adherence to surveillance shows room for amelioration [30].

Regarding therapeutic options, in recent years, major advances have been made in the field of systemic therapies [8]. Approved in 2007, the tyrosin-kinase inhibitor (TKI) sorafenib [31] was the first and only therapy for advanced HCC for almost one decade. With positive phase III studies for regorafenib in 2016 [32], cabozantinib [33] and lenvatinib in 2017 [34], as well as the VEGF-receptor 2 antagonist Ramucirumab in 2018 [35], in recent years, the therapeutic spectrum of first- and second-line therapies has expanded gradually, as well as the notably increased survival rates of several cancer entities. Thereby, the combination of VEGF antibody bevacizumab plus PD-L1 inhibitor atezolizumab was the first antibody-based therapy approved as first-line treatment for advanced HCC [36,37]. Several combinations including PD-L1 inhibitors or CTLA-4 antibodies are currently under investigation and immunotherapy is expected to extend the survival of patients with HCC in the years to come.

The augmented effect of novel systemic therapies might be reflected by the significantly improved 1-year survival rates in our study cohort, which is in line with rates previously reported in most other countries [38].

Analogous to HCC, varying incidence rates of cholangiocarcinomas worldwide may result from a different distribution of risk factors. Western and high-income countries are low-risk areas (incidences 0.35–2/100,000), while China and Thailand are high-risk areas with up to 40 times higher incidences [1]. Incidence rates of iCC show clearly rising trends in high-income countries, while those of eCC have only slightly increased or even remained stable [1,2,39]. In contrast to our previous analyses, where we observed rising incidences of iCC in Austria between 1990 and 2009 along with slightly decreasing numbers of eCC [11], we observed no significant dynamic in the current period between 2010 and 2018. These changing trends might be attributable to changes in the terminology. Adaptions of the International Classification of Diseases for Oncology (ICD-0) over the last decades, resulted in a significant bias concerning incidences of cholangiocarcinomas. In the first ICD-0 version there was no specific coding for Klatskin tumors (perihilar cholangiocarcinoma), in the second version a unique code was given and Klatskin tumors were classified as intrahepatic tumors, leading to an overestimation of iCC and underestimation of eCC [40]. In the ICD-0–3 version, it was cross-referenced to both iCC and eCC. Reported rising incidences of iCC and stable or decreasing incidences of eCC (e.g., USA and UK) might, thus, be substantially biased by these changes in the ICD-0 coding [41]. In the current analysis, the percentage of anatomically unclassified tumors was also significantly lower as compared with our previous data, which might implicate more robust information on incidence trends in recent years. This might account for the introduction of more diagnostic tools in recent years, such as albumin in situ hybridization assays [42]. The changing prevalences in the etiological landscape, as discussed above, might have also influenced cholangiocarcinoma in the same way as HCC, since both entities share the most common risk factors.

While age-adjusted mortality rates have remained stable over time in both iCC and eCC, in recent years, 1-year survival rates have been prolonged significantly. Surgery is the only curative treatment option for BTC. Unfortunately, the vast majority (60–70%) of cases are still diagnosed at advanced tumor stages, when tumor mass, infiltration, or spreading prohibits curative resection [1]. Published in 2010, the landmark study UK ABC-02 standardized cisplatin plus gemcitabine as first-line treatment for advanced BTCs worldwide [43]. As the only approved non-surgical therapeutic option for a long time, this combination therapy significantly improved OS.

Recently, the principle of targeted therapy has gained traction for cholangiocarcinoma. Based on specific targetable mutations (e.g., IDH1, FGFR2, VEGF, and NTRK) several (second-line) therapies have shown promising prolonged OS and have been approved, mainly for iCC [44,45]. Together with immunotherapy [44], survival rates are expected to further improve in the future. Recently, the TOPAZ-1 phase III study showed that combining cisplatin/gemcitabine with the PD-L1 inhibitor durvalumab further improved overall response rate (ORR), progression free survival (PFS), and median [46].

Regarding GBC, incidences are stable or decreasing worldwide [1]. Predisposing factors are higher age and female sex, lifestyle factors, environmental influences, as well as conditions causing chronic irritation and inflammation of the gallbladder. Cholelithiasis has been found to be the main risk factor, being present in 70–90% of all GBC cases [47]. Other diseases associated with an increased risk are primary sclerosing cholangitis, structural biliary tree abnormalities, choledochal cysts, obesity, or chronic infection with Salmonella typhi or Helicobacter bilis [1]. The formally observed trend of decreasing incidences and mortality rates in Austria further continued in women in our current study and is in line with previous data from most other countries. This international dynamic is most probably attributable to increasing routine cholecystectomy rates [48].

There has been a significant increase in OS over the three time periods. In most cases, the diagnosis of GBC is made either incidentally after cholecystectomy showing a rather good prognosis or in symptomatic patients at advanced stages with a poor prognosis. Since more than half of cases are diagnosed incidentally, increasing routine cholecystectomy rates seem to be the main reason for the better OS [48]. Therapeutic options at advanced stages are similar to those of cholangiocarcinoma and the standardization of cisplatin plus gemcitabine as first-line chemotherapy [43,45] has influenced the OS in GBC as well. Molecular-targeted therapies and immunotherapies are expected to further ameliorate the OS in the future [44].

Cancer of the ampulla of Vater is a rare entity and accounts for 0.2% of all gastrointestinal tumors with incidence rates of about 0.5 per 100,000 in Western countries. Due to the now, more frequent performance of routine upper GI endoscopy, the rate of AC detection has increased in recent years [1,49]. In Austria, we observed slightly fluctuating incidences without any significant dynamic.

Half of AC cases are eligible for surgical or endoscopic removal showing a good prognosis. Lymph node metastases are more frequently found as compared with other BTCs, which is in line with our data, and their presence often requires more radical surgical treatment.

At advanced tumor stages, chemotherapeutic options are the same as those for cholangiocarcinoma and GBC [45]. The prolonged OS in the most recent time period of our study can most probably be ascribed to advances in systemic treatment as well as to earlier detection rates due to the higher performance of routine endoscopies. Enhanced endoscopic treatment approaches, such as papillectomy and radio-frequency or argon plasma therapies together with refined stenting options to prevent biliary occlusion, might also have contributed to the improved outcomes.

The age-adjusted incidence rates in our study cohort differed markedly from those reported in a recent publication by Rumgay, H. et al. [50]. In this study, the age-adjusted incidence rates of primary liver cancer from 95 countries in 2018 were analyzed and pooled according to different world regions. Central Europe (including Austria) was pooled with Eastern Europe. However, both regions differ substantially in incidences of HCC and iCC. Furthermore, age-adjusted incidence rates were calculated based on the Segi–Doll world standard population, while the European Standard Population, 2013 was used in our study cohort. Thus, a direct comparison of absolute numbers of age-adjusted incidence rates is not possible [51].

A major strength of our study is the large number of included patients on a population level. Only 2% of our patients were classified as having a HBC of unspecified anatomical location and were excluded from the study cohort. Further, the close to 100% accuracy of follow-up data in the Austrian population registry make the OS data very robust.

Our study has some limitations. First, only data on OS were available for the comparison of the different time periods. Unfortunately, age-adjusted incidence and mortality rates per 100,000 were calculated based on the European Standard Population, 2013 for the period 2010–2018, whereas they were calculated based on the WHO Standard Population, 2001 in our previous studies [10,11]. Thus, a direct comparison of the absolute numbers of age-adjusted incidence rates and mortality rates between the time periods was not possible [51]. However, observed trends in incidences and mortalities as well as survival outcomes in both studies could be analyzed and compared. Second, similar to any analysis of this kind, a fraction of patients (21% in 2010–2018) were DCO cases, and therefore, could not be included in survival analyses. Third, data on lifestyle factors and anthropometric data including body mass index, dietary habits, and ethnicity were not available for this analysis.

## 5. Conclusions

In conclusion, while incidence rates have been decreasing further in female patients diagnosed with GBC, those of all other cancer entities have remained stable over time. The median OS significantly improved in all HBC subtypes as compared with those of the previous two decades, reflecting the impact of the improved diagnostic and therapeutic options. However, mortality rates have remained unchanged in almost all HBC entities, suggesting that patients are still diagnosed at advanced tumor stages, when curative treatments are no longer feasible. Thus, the surveillance of patients at risk (e.g., by biannual ultrasound screening in patients suffering from liver cirrhosis) should be further reinforced. Additionally, recently, advances have been made in the identification of biomarkers for the diagnosis of HBC. The intention is to establish a noninvasive diagnosis at very early tumor stages, for example, by the detection of circulating tumor DNA or micro-RNA for HCC or cholangiocarcinoma. Although not yet implemented in routine clinical practice, studies on biomarkers show very promising results, and might lead to an earlier diagnosis in HBC patients in the upcoming years [52,53].

## Figures and Tables

**Figure 1 cancers-14-03093-f001:**
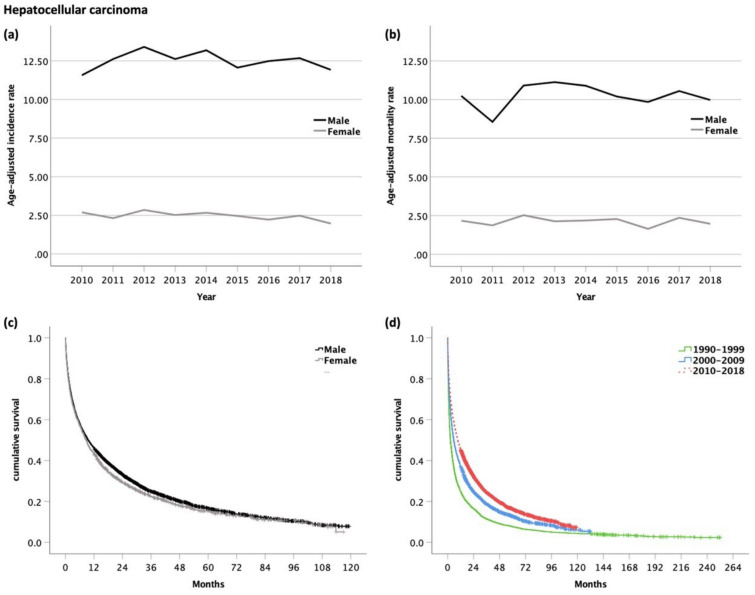
(**a**) Age-adjusted incidence rate (per 100,000); (**b**) age-adjusted mortality rate (per 100,000) of patients with hepatocellular carcinoma. Kaplan–Meier curves of patients with hepatocellular carcinoma: (**c**) Male vs. female patients (median OS for m and f, 9.5 and 6.8 months, respectively, *p* = 0.050); (**d**) comparison of OS with former time periods (median OS for 1990–1999, 2.6 months; 2000–2009, 5.6 months; 2010–2018, 9.3 months; *p* < 0.001).

**Figure 2 cancers-14-03093-f002:**
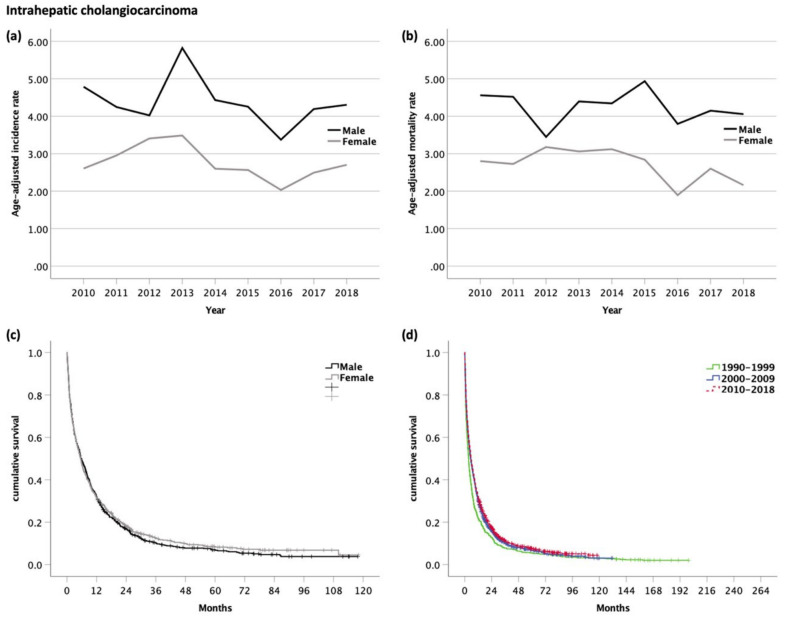
(**a**) Age-adjusted incidence rate (per 100,000); (**b**) age-adjusted mortality rate (per 100,000) of patients with intrahepatic cholangiocarcinoma. Kaplan–Meier curves of patients with intrahepatic cholangiocarcinoma: (**c**) Male vs. female patients (median OS for m and f, 5.7 and 5.5 months, respectively, *p* = 0.486); (**d**) comparison of OS with former time periods (median OS for 1990–1999, 3.5 months; 2000–2009, 5.3 months; and 2010–2018, 5.6 months; *p* < 0.001).

**Figure 3 cancers-14-03093-f003:**
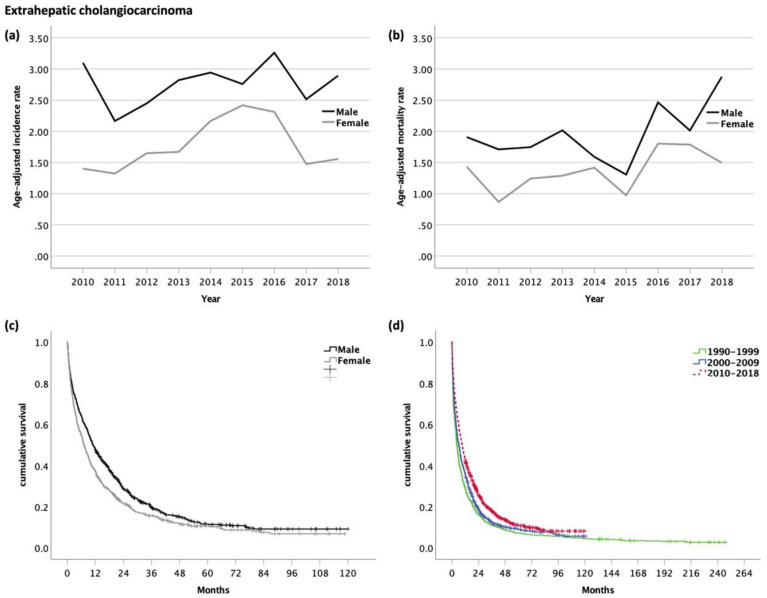
(**a**) Age-adjusted incidence rate (per 100,000); (**b**) age-adjusted mortality rate (per 100,000) of patients with extrahepatic cholangiocarcinoma. Kaplan–Meier curves of patients with extrahepatic cholangiocarcinoma: (**c**) Male vs. female patients (median OS for m and f, 10.1 and 7.1 months, respectively, *p* = 0.001); (**d**) comparison of OS with former time periods (median OS for 1990–1999, 4.4 months; 2000–2009, 6.3 months; and 2010–2018, 9.1 months; *p* < 0.001).

**Figure 4 cancers-14-03093-f004:**
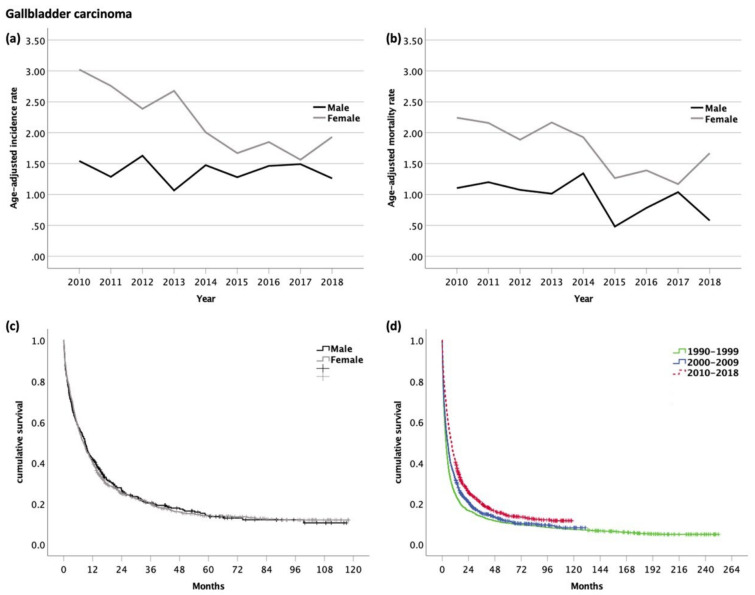
(**a**) Age-adjusted incidence rate (per 100,000); (**b**) age-adjusted mortality rate (per 100,000) of patients with gallbladder carcinoma. Kaplan–Meier curves of patients with gallbladder carcinoma: (**c**) Male vs. female patients (median OS for m and f, 8.8 and 8.2 months, respectively, *p* = 0.854); (**d**) comparison of OS with former time periods (median OS for 1990–1999, 3.4 months; 2000–2009, 4.9 months; 2010–2018, 3.4 8.5 months; *p* < 0.001).

**Figure 5 cancers-14-03093-f005:**
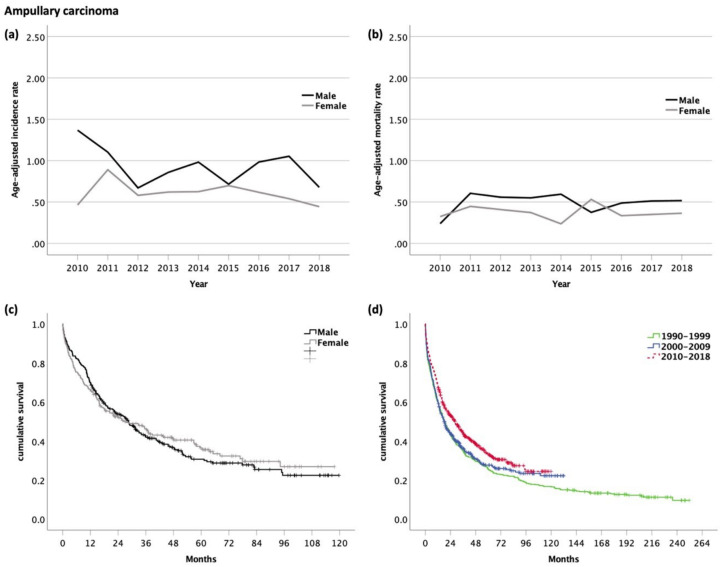
(**a**) Age-adjusted incidence rate (per 100,000); (**b**) age-adjusted mortality rate (per 100,000) of patients with ampullary carcinoma. Kaplan–Meier curves of patients with ampullary carcinoma: (**c**) Male vs. female patients (median OS for m and f, 28.5 and 27.5 months, respectively, *p* = 0.720); (**d**) comparison of OS with former time periods (median OS for 1990–1999, 18.3 months; 2000–2009, 18.2 months; and 2010–2018, 28.5 months; *p* < 0.001).

**Table 1 cancers-14-03093-t001:** Patient characteristics of the included five cancer entities.

		All Analyzed Patients (*n* = 12,577)	HCC (*n* = 7146)	iCC (*n* = 1858)	eCC (*n* = 1649)	GBC (*n* = 1365)	AC (*n* = 559)
Variable			*n* (%)	*n* (%)	*n* (%)	*n* (%)	*n* (%)
**Age**	Mean ± SD	71 ± 11.7	70 ± 11.5	71 ± 12.0	73 ± 11.4	71 ± 11.2	71 ± 12.2
**Sex**	Male	7959 (63)	5342 (75)	997 (54)	884 (54)	437 (32)	299 (54)
**(*n* (%))**	Female	4618 (37)	1804 (25)	861 (46)	765 (46)	928 (68)	260 (46)
**Metastases (*n* (%))**	Lymph node ^1^	1190 (10)	310 (4)	167(9)	289 (18)	255 (19)	169 (30)
	Distant ^2^	1489 (12)	574 (8)	412 (22)	214 (13)	343 (25)	43 (8)

Abbreviations: HCC, hepatocellular carcinoma; iCC, intrahepatic cholangiocarcinoma; eCC, extrahepatic cholangiocellular carcinoma; GBC, gallbladder carcinoma; AC, ampullary carcinoma. STD, standard deviation. ^1^ Not evaluable, *n* = 2629 (21%). ^2^ Not evaluable, *n* = 1755 (14%).

## Data Availability

Data on incidence (last updated in December 2021) were obtained from the Austrian National Cancer Registry (ANCR). All patients with hepatobiliary tumors in Austria are registered at the ANCR as notification is obliged by law (Cancer Statistics Act 1969 and Cancer Statistics Ordinance 2019). This is a population-based registry and has been operated by the National Statistical Institution (Statistics Austria) since 1969. For follow-up and to ascertain death certificate only (DCO) cases, ANCR data are linked with the official causes of death (CoD) statistics derived from Statistics Austria since 1983.

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
