# Peer review of "Changing Epidemiological Trends of Hepatobiliary Carcinomas in Austria 2010–2018"

_cancers, 2022, doi:10.3390/cancers14133093_

Round 1

Reviewer 1 Report

In the present original article, Hucke   et al. very nicely describe the trend in

 hepatobiliary carcinomas in Austria between 2010-2018. They compare their findings with their previously published work regarding their 2000-2009 data.

This work is very interesting. More information and revision of the text is required. Here are some suggestions.

Major:

1.     The authors compare the overall survival in each category of hepatobiliary carcinomas in the 2010-2018 period with 2000-2009 one. Is there any significant difference in the Age-adjusted incidence rate, too, between these two time periods? Is there an increase in incidence rate of HCC, iCC, eCC, etc? In the discussion, only the present period is discussed (Age-adjusted incidence and mortality rates remained stable in HCC, intra- and ex-trahepatic cholangiocarcinoma and ampullary carcinoma over time in both men and women.)

2.     How does the present study compare with the published one (PMID: 34942552) regarding HCC and iCC in the same European region? Please discuss.

3.     Expand the discussion relative to improvement in early diagnosis of HCC and BTCs.  Next-generation, non-invasive biomarkers are being searched for and are of great interest for diagnosis of HCC and BTCs when these are still asymptomatic (PMID: 34203269; PMID: 34779230; PMID: 35580963).

4.     DISCUSSION: Avoid describing results which have already been mentioned in the RESULTS section

5.     Is there any difference in the Age-adjusted incidence rate and overall survival if ethnicity is considered?

Minor:

1.       English should be thoroughly revised. There are too many mistakes, such as in the abstract: “Incidences were increasing over the last decades” should read “ Incidences increased over the last decades”. Please check text throughout (other example: are worldwide mostly rising, page 2). Punctuations not appropriately used

2.       Please check typographical errors: eg, (p=.0002), page 7; Salmonella typhi of Helicobacter bilis, page 12.

3.       Abstract should be written without too many abbreviations (too difficult to follow). GBC is used only in the abstract! Use only strictly essential abbreviations. Text is difficult to read as such, too many abbreviations to memorise. What is DCO? What is CCC in the sentence “In the last years the principle of targeted therapy was also gaining traction for CCC.” (page 12)?

4.       Avoid 1-sentence paragraphs, group sentences according to ideas/discussions through the text.

5.       Include page numbers.

6.       Cumulative survival curves in Figures 2-6 would be better appreciated in colour mode (like in Figure 1).

7.       Figure 1: Overview (last scheme)…label it Figure 7.

Reviewer 2 Report

Hucke et. al., in their paper provided an updated epidemiological trend of HCC in Austria for the period of 2010-2018 and compared it with their previous repot. The study included the incidence, mortality and survival trends. I have only few minor comments to make:

·        The full form of abbreviation DCO, OS, CI, NAFLD, NASH is missing

·        The authors talk about drop in alcohol consumption, however eating/drinking habits or any exercise routines were not included in the paper Were these variables included in the study?

·        The authors are suggested to improve their conclusion and how this study can be beneficial to the researchers and clinicians working in the field of hepatobiliary carcinomas.

Reviewer 3 Report

The current article investigated the incidence, mortality, and survival trends of hepatobiliary cancer (HBC) over the last decade in Austria. The authors report that while age-adjusted incidence rates remained stable in almost all HBC types, except in gall-bladder cancer, overall survival significantly improved. This research provides important information on the epidemiologic trend of HBC. Overall, this is a well-written manuscript of a well-conducted study. The methodology is sound, leading to an appropriate conclusion. The data provided support the conclusion.

I have a few minor suggestions:

1.    Consistency of terms used: Please keep the terms consistent throughout. HBC has been referred to as primary liver cancer (in simple summary) and sometimes hepatobiliary tumors.

2.    The discussion is too long, and not all of it is relevant for the current study. For example, a detailed description of HCC treatment is not needed. Some of the historical facts about the treatment of various HBC subtypes should be shortened. The same goes for cholangiocarcinoma treatment.

3.    The authors should include a paragraph describing the limitations of the study.

4.    The authors should provide detailed information about the Austrian national database. Is it an electronic database? Are the data stored and pooled with the help of any specific software? Was any portion of the data manually extracted from other sources, like death certificates?

5.    Patient characteristics: Table 1 provides mean age- why not median age and the range?

6.    Figure legend of figure 1 (1st line) mentions ‘primary liver and biliary tract cancer’- are the authors referring to HBC (consistency of terms should be maintained).

Round 2

Reviewer 1 Report

Authors have responded to this reviewer's query.